# Quanti-Qualitative Response of Swiss Chard (*Beta vulgaris* L. var. *cycla*) to Soil Amendment with Biochar-Compost Mixtures

**Angela Libutti** [1,*] and **Anna Rita Rivelli** [2,*]

1    Department of Science of Agriculture, Food, Natural Resources and Engineering, University of Foggia, Via Napoli, 25, 71122 Foggia, Italy
2    School of Agricultural, Forest, Food and Environmental Sciences, University of Basilicata, Via dell'Ateneo Lucano, 10, 85100 Potenza, Italy
*    Correspondence: angela.libutti@unifg.it (A.L.); annarita.rivelli@unibas.it (A.R.R.)

**Abstract:** In recent years, soil addition with organic amendments, such as biochar and compost, has gained attention as an effective agronomic practice to sustain soil fertility, enhance plant growth and crop yield. Well known are the positive effects of compost on yield of a wide crop varieties, while both positive and negative responses are reported for biochar Therefore, the aim of the study was to verify the effect of biochar mixed with three types of compost on quanti-qualitative response of Swiss chard (*Beta vulgaris* L. *cycla*), a leafy green vegetable rich in dietary antioxidants, largely consumed worldwide. A factorial experiment in pots with two factors, including biochar (without biochar and with biochar from vine pruning residues) and compost (without compost, with compost from olive pomace, with vermicompost from cattle manure, and with compost from cattle anaerobic digestate), was setup. Two growth cycles were considered, and a set of quantitative (height of plants, number, area and fresh weight of leaves) and qualitative parameters (carotenoids, chlorophyll, total N, and $NO_3^-$ content of leaves) were analyzed. Biochar decreased plant growth and $NO_3^-$ leaf content; on the contrary, it increased total N leaf content, while compost improved all the considered parameters. The interactive effect of biochar and compost was evident only on total N and $NO_3^-$ leaf content. In our experimental conditions, the compost showed to be the best option to improve Swiss chard growth and increase the content of phytopigments, while the biochar-compost mixtures did not produce the expected effect.

**Keywords:** biochar; compost; vermicompost; biochar-compost mixtures; soil fertility; Swiss chard; nitrate content; leaf pigment content

## 1. Introduction

Over the last half century, chemical-based fertilizers have played a significant role in increasing crop yield, but the over-application, especially N fertilizers, has caused soil quality degradation and other serious environmental impacts [1]. The application of organic residues to the soils, by ensuring the biological cycling of nutrients [2], could contribute to restore soil fertility. This is recognized as a sustainable strategy of soil management and agricultural productivity and is also consistent with the "circular economy" principle of "closing the loop".

In this regard, there is a growing interest in the use of valuable and safe products deriving from technologies of organic wastes and residues recycling, such as biochar and compost. Biochar is a carbon-rich, porous, low-density material obtained by thermal treatments in oxygen-limited environment (pyrolysis or gasification) of different feedstock (manure, sewage sludge, forest and agriculture residues) [3–5]. Compost is the humus-like product of the decomposition, stabilization, and sanitation of organic residues deriving from plants and animals, through the action of aerobic microorganisms under controlled conditions [6]. A number of environmental and agronomic benefits deriving from compost and biochar have been documented in several reviews and meta-analyses studies [7,8].

These organic amendments are reported to mitigate climate changes by atmospheric $CO_2$ sequestration, to improve soil physical (aggregation, density, root penetration) and chemical (pH and CEC) properties, water-holding capacity and nutrient retention, organic matter and nutrient cycling, to stimulate soil microbial, microfauna, and mesofauna communities, to reduce nutrient leaching and increase heavy-metal sequestration.

Well known are the positive effects of compost on yield of a wide crop varieties [3], while both positive and negative crop yield responses are reported for biochar [7–10]. According to several authors [11], combining biochar with compost might minimize or even prevent negative effects of biochar on plants. Biochar-compost blend could take advantage from bringing together the physico-chemical characteristics of the two amendments, such as the biochar ability to increase water holding capacity, retain nutrients, stimulate soil microorganisms, and the compost capacity to provide labile organic carbon to the soil and nutrients to the plants [12]. For these reasons, biochar-compost application has been suggested as a soil management approach able to improve soil quality and crop yield [13]. In particular, Agegnehu et al. [8] reported that the application of a biochar-compost combination is more effective in improving soil properties and yields of field and horticulture crops than biochar alone.

Among the horticulture crops suitable for the cultivation with organic amendments is included the Swiss chard (*Beta vulgaris* L. *cycla*), a leafy green vegetable widely cultivated in many temperate regions of the world and largely consumed fresh, frozen, or canned. The species is a rich source of phytopigments (chlorophyll and carotenoids) and dietary antioxidants (flavonoids and phenolic compounds) with free-radical scavenging capacity that play an important role in reducing the risk of chronic and neurodegenerative diseases [14]. However, Swiss chard is among the leafy vegetables accumulating excessive nitrate amount, with over 60% in the petiole [15–17]. Several studies [18,19] showed that high N fertilization levels negatively affect the quality of Swiss chard leaves, instead Razgallah et al. [20] found lower nitrate content in organically than conventionally cultivated Swiss chard. In this context, the application of organic amendments to Swiss chard cultivation, could be a way to reduce or even replace the inorganic fertilization, while assuring plant nutrition, crop yield, and quality.

Some experimental evidences reported higher Swiss chard growth and yield using vermicompost [21] or compost [22]. Studies about the application of biochar-compost mixtures to Swiss chard have not been evaluated and are not documented in literature. Our previous paper [23] focused on Swiss chard response to biochar (from vine pruning residues) and different compost types (from olive pomace or cattle manure or cattle anaerobic digestate with wheat straw or cattle anaerobic digestate with crop residues and wheat straw), each added to the soil in two doses, in order to provide 140 and 280 kg N ha$^{-1}$. Swiss chard responded positively to composts, particularly to those from animal wastes and to the higher N dose, showing a higher yield and a better product quality, while biochar did not lead to positive or negative effects. Following these results and using the same types of organic amendments, the aim of the present study was to test whether mixing biochar with composts could be a more powerful strategy to enhance biochar effect on plant growth and qualitative characteristics of product, in a perspective of sustainable agriculture.

To this purpose a pot experiment was carried out on plants grown on soil amended with biochar (from vine pruning residues), three composts (a compost from olive pomace, a vermicompost from cattle manure and a compost from cattle anaerobic digestate with crop residues and wheat straw), and biochar-compost mixtures.

## 2. Materials and Methods

### 2.1. Experimental Design and Plant Growing Conditions

An experiment on Swiss chard (*Beta vulgaris* L. var. *cycla*) was conducted during 2017 at the greenhouse of the University of Basilicata (South Italy), in Potenza (PZ, 40°38′ N–15°48′ E, 819 m a.s.l.).

The plants were grown in pots and the experimental design included the following two factors: (1) biochar (B) (without biochar, B–; with biochar from vine pruning residues, B+); (2) compost (C) (without compost, C–; with compost from olive pomace, $C_{OP}$+; with vermicompost from cattle manure, $C_W$+; with compost from 79% cattle anaerobic digestate, 11% crop residues, and 10% wheat straw, $C_D$+). The combination of the two experimental factors, biochar and compost, resulted in the following eight experimental treatments to the soil:

(1)　Without biochar and without compost as a control (B– C–);
(2)　With biochar and without compost (B+ C–);
(3)　Without biochar and with compost from olive pomace (B– $C_{OP}$+);
(4)　With biochar and with compost from olive pomace (B+ $C_{OP}$+);
(5)　Without biochar and with vermicompost (B– $C_W$+);
(6)　With biochar and with vermicompost (B+ $C_W$+);
(7)　Without biochar and with compost from cattle anaerobic digestate (B– $C_D$+);
(8)　With biochar and with compost from cattle anaerobic digestate (B+ $C_D$+).

Based on the findings of our previous study [23], biochar and composts were added to the soil in order to provide 280 kg N ha$^{-1}$, both when used alone (B+ C–, B– $C_{OP}$+, B– $C_W$+, B– $C_D$+) or in mixture (B+ $C_{OP}$+, B+ $C_W$+, B+ $C_D$+). In the latter case, biochar and composts were mixed, having an N loading ratio of 50:50. Considering the total nitrogen content of biochar and composts and assuming an application within the 0.15 m top layer of a soil having a bulk density of 1.3 Mg m$^{-3}$, the dose of 280 kg N ha$^{-1}$ was equivalent to 20, 19, 24, and 26 Mg ha$^{-1}$ as fresh matter of biochar, compost from olive pomace, vermicompost, and compost from cattle anaerobic digestate, respectively.

The experimental soil was collected from the topsoil (0–20 cm) of a field located in the agricultural area of Potenza district (Southern Italy). Before use, it was air-dried and passed through a 2-mm sieve. The composts were purchased at the industrial plants of Eboli (Salerno district, Italy) and Montescaglioso (Matera district, Italy), respectively. The biochar was produced at the STAR*Facility Center of Foggia University (Foggia, Southern Italy), by pyrolysis of vine (*Vitis vinifera* L.) pruning residues, at a reaction temperature of 650 °C and 8 h residence time in a pilot-scale fixed bed tubular reactor (30 L capacity). Before use, biochar was ground and passed through a 2 mm sieve.

Plastic pots (13 × 13 × 24 cm) were used for the experiment. They were prepared two months prior plant transplanting, by firstly placing 2-cm layer of expanded clay at the bottom to improve water drainage and then adding 2 kg of untreated or treated soil. The latter was obtained by, respectively, adding to the soil biochar, composts, or biochar-compost mixtures and homogeneously mixing. A randomized complete block design with the eight above-described treatments, each replicated four times, was set up for a total of 32 pots.

A single Swiss chard seedling was transplanted into each pot and soil surface covered by a polyethylene beads layer of 3 cm to minimize evaporation. Plants were watered every 2–3 days by applying a water volume of 120 mL at each irrigation. Two growth cycles, each of about four weeks, were considered, and two cuts of marketable leaves were performed. The first leaf cut was made at 34 days after transplanting (DAT), at the end of the first growth cycle, avoiding to blind the plant thus promoting the development of the basal leaflets newly formed. The second leaf cut was made at 60 DAT, at the end of the second growth cycle.

### 2.2. Soil and Organic Amendments Analysis

Before pot experiment started, soil samples were air-dried, crushed, passed through a 2-mm sieve, and analyzed for the set of physico-chemical characteristics reported in Table 1.

**Table 1.** Main physico-chemical properties of the soil utilized in the experiment.

| Property | Unit | Value |
|---|---|---|
| Clay | % | $22.4 \pm 0.7$ |
| Silt | % | $11.5 \pm 0.8$ |
| Sand | % | $66.1 \pm 0.8$ |
| pH | - | $7.5 \pm 0.1$ |
| EC | $dS\,m^{-1}$ | $0.4 \pm 0.1$ |
| $P_2O_5$ | $mg\,kg^{-1}$ | $28.0 \pm 0.6$ |
| $C_{org}$ | $g\,kg^{-1}$ | $7.9 \pm 0.7$ |
| OM | % | $1.4 \pm 0.7$ |
| C/N | - | $7.2 \pm 0.8$ |
| Total N | ‰ | $1.1 \pm 0.5$ |
| $NO_3^-$ | $mg\,kg^{-1}$ | $48.0 \pm 0.3$ |
| $Na^+$ | $mg\,kg^{-1}$ | $25.0 \pm 0.4$ |
| $Ca^{2+}$ | $mg\,kg^{-1}$ | $3289.0 \pm 0.9$ |
| $Mg^{2+}$ | $mg\,kg^{-1}$ | $215.0 \pm 0.9$ |
| $K^+$ | $mg\,kg^{-1}$ | $368.0 \pm 0.8$ |

Values are means ($n = 3$) $\pm$ standard errors.

In particular, the size distribution of the mineral particles was determined by the pipette-gravimetric method. The pH was determined in the extract of 1:2.5 (*w/v*) soil/water suspension by a digital pH meter (GLP 22 pH-meter, Crison Instruments, Alella, Barcelona) and the electrical conductivity (EC) in the extract of saturated soil paste by a digital conductivity meter (GLP 31 EC-meter, Crison Instruments, Barcelona). The available phosphorus ($P_2O_5$) was determined by extraction with sodium bicarbonate [24], the organic carbon by the dichromate oxidation method [25], and then converted to organic matter (OM) by the conventional factor 1.724. The cation ($Na^+$, $Ca^{2+}$, $Mg^{2+}$, and $K^+$) content was determined in the extract of soil saturated paste by an atomic absorption spectrometer (AAS 2380, Perkin-Elmer, Seer Green, Beaconsfield, Buckinghamshire, UK), the total nitrogen (total N) by the Kjeldahl method [26] and the nitrate ($NO_3$-) by spectrophotometric analysis after extraction with 2 M KCl from the soil [27]. The soil was found to be neutral (pH = 7.5), relatively low in OM (1.4%), and to have a medium content of total N (1.1‰) and a sandy-clay-loam texture (clay, 22.4%; silt, 11.5%; sand, 66.1%). Further information about the main soil characteristics are reported in Table 1.

Samples of biochar and composts were analyzed for the main physico-chemical properties showed in Table 2.

The pH and electrical conductivity (EC) were determined in the extract of 1:20 (*w/v*) organic amendment/water suspension, after shaking the suspension and waiting an equilibrium time of 90 min, by a digital pH-meter (GLP 22+ pH-meter, Crison Instruments, Barcelona) and a digital conductivity meter (GLP 31+ EC-meter, Crison Instruments, Barcelona), respectively. The fixed carbon, volatile solids, ash and moisture (proximate analysis) were determined by a Thermogravimetric - TGA Analyzer (LECO-TGA701), the carbon (C), nitrogen (N), hydrogen (H) and organic carbon ($C_{org}$) (ultimate analysis) by a CHN (Carbon, Nitrogen, Hydrogen) Elemental Analyzer (CHN LECO628). The carbon to nitrogen ratio (C/N) was also calculated for each of the organic amendments used. Only for the biochar, sulfur (S) was determined using a S module (S LECO628) combined with the CHN Elemental Analyzer and oxygen (O) was calculated by the difference: O (%) = 100-C-H-N-S-ash. Finally, carbon stability of biochar was evaluated indirectly by the molar ratios of hydrogen to organic carbon ($H/C_{org}$) and oxygen to organic carbon ($O/C_{org}$).

**Table 2.** Main chemical properties of the organic amendments utilized in the experiment: biochar from vine pruning residues (B); compost from olive pomace (COP); vermicompost from cattle manure (CW); compost from 79% cattle anaerobic digestate, 11% crop residues and 10% wheat straw (CD).

| Property | Unit | Organic Amendment | | | |
|---|---|---|---|---|---|
| | | **B** | **COP** | **CW** | **CD** |
| pH | - | $11.3 \pm 0.1$ | $7.9 \pm 0.0$ | $7.6 \pm 0.1$ | $8.7 \pm 0.0$ |
| EC | dS m$^{-1}$ | $3.6 \pm 0.2$ | $2.4 \pm 0.1$ | $2.7 \pm 0.0$ | $2.0 \pm 0.0$ |
| Fixed carbon | % | $69.8 \pm 0.0$ | $3.9 \pm 0.1$ | $0.5 \pm 0.0$ | $2.4 \pm 0.0$ |
| Volatile solids | % | $17.0 \pm 0.0$ | $53.9 \pm 0.0$ | $32.34 \pm 0.1$ | $69.8 \pm 0.0$ |
| Ash | % | $13.3 \pm 0.0$ | $42.2 \pm 0.1$ | $67.1 \pm 0.1$ | $27.9 \pm 0.1$ |
| Moisture | % | $5.2 \pm 0.0$ | $28.8 \pm 0.1$ | $35.5 \pm 0.1$ | $34.0 \pm 0.1$ |
| C | % | $67.7 \pm 0.3$ | $60.9 \pm 1.2$ | $25.2 \pm 0.1$ | $34.5 \pm 0.3$ |
| H | % | $2.4 \pm 0.1$ | $4.6 \pm 0.2$ | $1.4 \pm 0.2$ | $4.4 \pm 0.1$ |
| N | % | $1.5 \pm 0.0$ | $2.6 \pm 0.1$ | $1.6 \pm 0.0$ | $2.6 \pm 0.0$ |
| C$_{org}$ | % | $66.7 \pm 0.0$ | $56.3 \pm 0.1$ | $20.1 \pm 0.2$ | $26.6 \pm 0.6$ |
| C/N | - | $45.1 \pm 0.8$ | $21.6 \pm 0.8$ | $12.4 \pm 0.1$ | $10.3 \pm 0.3$ |
| S | % | $0.0 \pm 0.0$ | - | - | - |
| H/C$_{org}$ | - | $0.4 \pm 0.0$ | - | - | - |
| O/C$_{org}$ | - | $0.4 \pm 0.0$ | - | - | - |

Values are means ($n$ = 3) $\pm$ standard errors.

As reported in Table 2, the biochar (B) was characterized by the typical alkaline pH observed by other authors [28] and resulted in an EC value higher than the soil. The C content was well within the threshold fixed by the European Biochar Certificate [29] and the C$_{org}$ content according to the Class 1 defined by the International Biochar Initiative (IBI) Standard [30]. The H/C$_{org}$ molar ratio was found to comply with the requirements both of European Biochar Certificate (EBC) and IBI Standard (H/C$_{org} \leq 0.7$). Lower than 0.7 H/C$_{org}$ molar ratios indicate biochar long-term stability, persistence in the soil, and contribution to soil carbon sequestration [31], while higher than 0.7 H/C$_{org}$ molar ratios indicate a non-pyrolytic biochar or deficiencies of pyrolysis process [32]. Similarly, the O/C$_{org}$ ratio, which is likewise relevant for characterizing biochar and differentiating it from other carbonization products [32], was found to meet the EBC and IBI-Standard requirements (O/C$_{org} \leq 0.4$).

Composts resulted to be mature and stable products. They showed a neutral or slightly alkaline pH and a higher EC value than the receiving soil. The two composts, respectively, from olive pomace and cattle anaerobic digestate, crop residues, and wheat straw were characterized by a higher total N content than vermicompost from cattle manure. Moreover, compost from olive pomace was found to have a very high C$_{org}$ content, which was only slightly below C$_{org}$ content of biochar and more than double of C$_{org}$ content of the other two composts.

### 2.3. Plant Growth Analysis

Starting from 10 days after transplanting, a counting of leaf number (LN) was carried out twice per week over the two growth cycles until leaf cutting. In addition, at the end of the two growth cycles, plant height (H), leaf area (LA), and leaf fresh weight (FW) were also recorded. H was determined before leaf cutting, by measuring the plant from the soil level to the top of the longest leaf. LA and FW were determined after leaf cutting. To this purpose, the leaves were weighed by using an analytical balance to obtain FW and scanned by using a LI-COR leaf area meter (Model 3100, Inc., Lincoln, NE, USA) to obtain LA.

### 2.4. Pigment, Total Nitrogen, and Nitrate Leaf Content Analysis

At the end of the first growth cycle, before leaf cutting, the chlorophyll content of the leaves was estimated by using a handheld Soil Plant Analysis Development - SPAD 502 m (Konica-Minolta corporation, Ltd., Osaka, Japan). Average SPAD meter values were

calculated from three readings taken from the tip to the base of the youngest and fully expanded leaf per plant (SPADleaf) and all the leaves per plant (SPADplant).

After leaf cutting, a sample of fresh leaf tissue was collected and analyzed for carotenoids (CA), chlorophyll a (CHLa), and chlorophyll b (CHLb) contents. The fresh leaf tissue was ground in 2 mL 80% acetone (*v/v*) by using a glass homogenizer, then the homogenate was centrifuged at $2300\times$ rpm for 2 min and finally the leaf extract was spectrophotometrically analyzed. The CA, CHLa, and CHLb contents were calculated from absorbances at 480 nm [33], 646.6, and 663.6 [34], respectively, and the total chlorophyll (CHL) content was obtained as the sum of CHLa and CHLb. From pigment contents, the ratios CHLa/CHLb and CHL/CA were calculated. After the analytical determinations, CHL values were correlated with SPAD meter values. The correlation resulted highly significant ($R^2 = 0.88$) (data not shown); therefore, the SPAD meter was used as a rapid and non-destructive method to indirectly obtain an accurate estimate of the total leaf chlorophyll content, during the second Swiss chard growth cycle.

The total nitrogen (total N) and nitrate ($NO_3^-$) content of Swiss chard leaves were analyzed on dried leaf tissues, after heath treatment at a temperature of $70°$ C in a drying oven, respectively, by the Kjeldahl method [26] and the colorimetric method based on nitration of salicylic acid [35].

### 2.5. Statistical Analysis

All experimental data were tested for differences using analysis of variance (ANOVA) following a two-factor randomized complete block design. The dataset was preliminary tested for the normal distribution of the experimental error and the common variance of the experimental error, by applying the Shapiro–Wilk and Bartlett's tests, respectively.

A two-way ANOVA was carried out to examine the effect of the factors, biochar (B, two levels: without biochar and with biochar from vine pruning residues) and compost (C, four levels: without compost, with compost from olive pomace, with vermicompost from cattle manure, and with compost from cattle anaerobic digestate, crop residues, and wheat straw) and their interaction (C $\times$ B). The model was: $y_{i,j,k} = \mu + \gamma_k + \alpha_i + \beta_j + \alpha\beta_{i,j} + \varepsilon_{i,j,k}$, where $\mu$ is the overall mean; $\gamma$ is the effect of block; $\alpha$ is the effect of biochar; $\beta$ is the effect of compost; $\alpha\beta$ is the interactive effect of biochar and compost factors; $\varepsilon$ is the error with mean 0 and standard deviation $\sigma$.

From the ANOVA results, if the two main effects were significant with $p \leq 0.05$, Student's t test for biochar factor or Tukey honestly significant difference test for compost factor were applied; if the interactive effect was significant with $p \leq 0.05$, Tukey honestly significant difference post hoc test was applied to conduct pairwise comparisons and classify the eight experimental treatments.

All the statistical analyses were performed using the JMP software package, version 11 (SAS Institute Inc, Cary, NC, USA).

## 3. Results

### 3.1. Plant Growth

The leaf number (LN) produced by Swiss chard plants in the course of the experiment is reported in Figure 1. Analysis of variance revealed that neither of the two experimental factors, biochar and compost, nor their interaction, biochar x compost, influenced the LN values determined twice per week over the first and second growth cycle, and at the first and second cut, respectively, performed at 34 and 60 DAT.

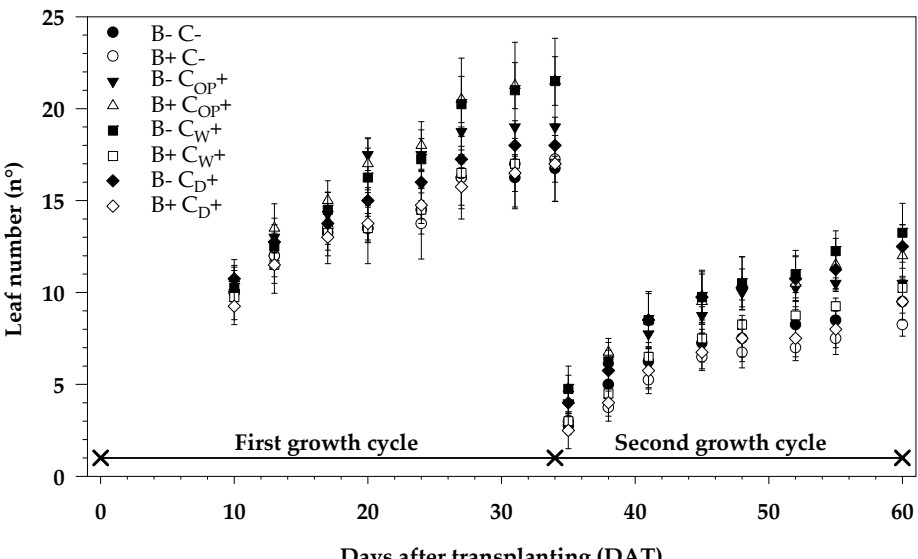

**Figure 1.** Leaf number counted twice per week over the first and second growth cycle, and at the first and second cut, respectively, performed at 34 and 60 days after transplanting (DAT), on Swiss chard plants grown on the following eight experimental treatments: B– C–, without biochar and without compost as a control; B+ C–, with biochar and without compost; B– $C_{OP}$+, without biochar and with compost from olive pomace; B+ $C_{OP}$+, with biochar and with compost from olive pomace; B– $C_W$+, without biochar and with vermicompost; B+ $C_W$+, with biochar and with vermicompost; B– $C_D$+, without biochar and with compost from cattle anaerobic digestate; B+ $C_D$+, with biochar and with compost from cattle anaerobic digestate. Values are means (*n* = 4) ± standard errors (vertical bars).

Overall, LN increased over time, always showing a higher value in the course of the first growth cycle than the second one. Both at the first and the second cut, a lower LN value was showed by plants grown on soil treated with biochar (B+) than without biochar (B–). On the contrary, 19 and 32% higher LN values were, respectively, detected on plants treated with compost from olive pomace ($C_{OP}$+) at the first cut and vermicompost ($C_W$+) at the second one, than C– (without compost). As average of the eight experimental treatments, LN values of 18.3 and 10.7 were detected at the first and second cut, respectively.

All the other growth parameters measured on Swiss chard plants at the end of the two growth cycles are shown in Table 3.

At the first leaf cut (Table 3), plant height (H), leaf area (LA), and leaf fresh weight (FW) were significantly influenced by biochar (*p* ≤ 0.001 for H and FW, *p* ≤ 0.01 for LA) and compost (*p* ≤ 0.001). When plants were grown on soil added with biochar (B+), lower values of H, LA, and FW, by 11, 15, and 24%, respectively, were observed than without biochar (B–). On the contrary, higher H, LA, and FW values were shown by plants grown on soil treated with the two composts from animal wastes, i.e., vermicompost from cattle manure ($C_W$+) and compost from cattle anaerobic digestate ($C_D$+), than with compost from olive pomace ($C_{OP}$+) that did not differ from no compost addition (C–).

At the second leaf cut (Table 3), H significantly differed among the composts m and was significantly influenced also by compost x biochar interaction (*p* ≤ 0.05). In detail, plants grown on soil added with vermicompost ($C_W$+) showed a 20% higher H value than without compost (C–) and with compost from olive pomace ($C_{OP}$+) that did not differ each other. Moreover, H reached the highest value in B– $C_D$+ treatment (without biochar and with compost from cattle anaerobic digestate) and the lowest in B+ $C_D$+ (with biochar and compost from cattle anaerobic digestate), even if not significantly different from B+ $C_{OP}$+ (with biochar and compost from olive pomace). LA and FW were significantly influenced by both biochar (*p* ≤ 0.05 for LA, *p* ≤ 0.001 for FW) and compost (*p* ≤ 0.01 for LA, *p* ≤ 0.001 for FW) (Table 3). As already observed at the first leaf cut, also at the second one, soil addition with biochar (B+) determined lower LA and FW values, by 14 and 23%,

respectively, than without biochar (B–). Furthermore, plants grown on soil treated with vermicompost ($C_W+$) and compost from cattle anaerobic digestate ($C_D+$) showed higher LA values than without compost (C–), accounting, on average, for an increase of 34%. FW values were higher on $C_W+$ and $C_D+$ treated plants than $C_{OP}+$ that did not differ from C–, accounting, on average, for a 15% increase.

**Table 3.** Effect of biochar, compost and biochar x compost interaction on plant height (H), leaf area (LA), and leaf fresh weigh (FW), at the first and second cut of Swiss chard.

| Experimental Factor | First Cut | | | Second Cut | | |
|---|---|---|---|---|---|---|
| | **H** cm | **LA** cm$^2$ | **FW** g | **H** cm | **LA** cm$^2$ | **FW** g |
| *Biochar (B)* | | | | | | |
| **B–** | 12.3 ± 0.3 a | 152.0 ± 8.0 a | 8.4 ± 0.4 a | 9.5 ± 0.3 | 74.0 ± 4.4 a | 4.1 ± 0.3 a |
| **B+** | 10.9 ± 0.4 b | 128.9 ± 7.8 b | 6.4 ± 0.5 b | 8.8 ± 0.3 | 63.8 ± 4.2 b | 3.1 ± 0.2 b |
| *Compost (C)* | | | | | | |
| **C–** | 10.1 ± 0.3 b | 110.2 ± 6.9 b | 5.9 ± 0.5 b | 8.5 ± 0.4 b | 57.2 ± 2.8 b | 2.9 ± 0.1 b |
| **$C_{OP}+$** | 11.0 ± 0.5 b | 128.9 ± 8.6 b | 6.6 ± 0.5 b | 8.5 ± 0.3 b | 65.3 ± 2.9 ab | 3.2 ± 0.2 b |
| **$C_W+$** | 13.3 ± 0.3 a | 165.3 ± 7.5 a | 8.8 ± 0.6 a | 10.2 ± 0.4 a | 80.2 ± 6.1 a | 4.3 ± 0.3 a |
| **$C_D+$** | 12.0 ± 0.5 a | 157.4 ± 9.0 a | 8.3 ± 0.5 a | 9.3 ± 0.5 ab | 73.0 ± 5.6 a | 4.1 ± 0.3 a |
| *Biochar x Compost* | | | | | | |
| **B– C–** | 10.6 ± 0.3 | 120.3 ± 4.8 | 6.7 ± 0.6 | 8.4 ± 0.6 ab | 56.2 ± 4.7 | 3.0 ± 0.2 |
| **B+ C–** | 9.7 ± 0.4 | 100.0 ± 11.5 | 5.1 ± 0.4 | 8.6 ± 0.4 ab | 58.3 ± 3.5 | 2.7 ± 0.2 |
| **B– $C_{OP}+$** | 12.1 ± 0.6 | 140.8 ± 4.5 | 7.6 ± 0.6 | 8.8 ± 0.7 ab | 67.2 ± 5.0 | 3.5 ± 0.2 |
| **B+ $C_{OP}+$** | 9.8 ± 0.6 | 117.1 ± 9.0 | 5.5 ± 0.5 | 8.2 ± 0.3 b | 63.3 ± 2.4 | 2.8 ± 0.2 |
| **B– $C_W+$** | 13.7 ± 0.6 | 178.9 ± 7.7 | 10.1 ± 0.7 | 10.1 ± 0.3 ab | 86.6 ± 5.0 | 4.9 ± 0.2 |
| **B+ $C_W+$** | 12.8 ± 0.8 | 151.6 ± 20.6 | 7.6 ± 1.0 | 10.3 ± 0.5 ab | 73.7 ± 4.3 | 3.6 ± 0.3 |
| **B– $C_D+$** | 12.7 ± 0.4 | 168.0 ± 10.7 | 9.2 ± 0.4 | 10.5 ± 0.3 a | 86.0 ± 3.3 | 4.8 ± 0.2 |
| **B+ $C_D+$** | 11.4 ± 0.7 | 146.9 ± 16.5 | 7.4 ± 0.7 | 8.1 ± 0.5 b | 59.9 ± 4.0 | 3.3 ± 0.3 |
| *Significance* | | | | | | |
| **B** | *** | ** | *** | ns | * | *** |
| **C** | *** | *** | *** | ** | ** | *** |
| **B x C** | ns | ns | ns | * | ns | ns |

Values are means ($n$ = 4) ± standard errors. In each column, means followed by the same letters are not significantly different ($p \leq 0.05$; Tukey's test). *, F test significant at $p \leq 0.05$; **, F test significant at $p \leq 0.01$; ***, F test significant at $p \leq 0.001$; ns, not significant. B, biochar; C, compost; $C_{OP}$, compost from olive pomace; $C_W$, vermicompost; $C_D$, compost from cattle anaerobic digestate; – and +, without and with, respectively.

### 3.2. Pigment Leaf Content

The carotenoids (CA), chlorophyll a (CHLa), chlorophyll b (CHLb), and total chlorophyll (CHL) leaf contents measured at the first cut (Table 4) resulted statistically different only among the composts ($p \leq 0.01$, $p \leq 0.01$, $p \leq 0.05$, and $p \leq 0.01$ for CA, CHLa, CHLb, and CHL, respectively).

Higher CA, CHLa, CHLb, and CHL contents were measured on plants treated with compost from cattle anaerobic digestate ($C_D+$) than without compost (C–), showing increases in leaf tissues equal to 29, 37, 36, and 37%, respectively.

The chlorophyll a to chlorophyll b ratio (CHLa/CHLb) (Table 4) was significantly affected only by biochar ($p \leq 0.01$). In particular, higher CHLa/CHLb value was detected in plants grown on soil treated with biochar (B+) than without biochar (B–), accounting for a 5% increase.

On the total chlorophyll to carotenoids ratio (CHL/CA) (Table 4), only the main effect of compost ($p \leq 0.05$) resulted as statistically significant. The highest CHL/CA value was detected on plants treated with compost from olive pomace ($C_{OP}+$), while the lowest in C– (without compost), accounting for a 7% increase.

**Table 4.** Effect of biochar, compost, and biochar x compost interaction on carotenoids (CA), chlorophyll a (CHLa), chlorophyll b (CHLb), total chlorophyll (CHL) leaf content, chlorophyll a to chlorophyll b (CHLa/CHLb) and total chlorophyll to carotenoids (CHL/CA) ratios of Swiss chard.

| Experimental Factor | CA | CHLa | CHLb | CHL | CHLa/CHLb | CHL/CA |
|---|---|---|---|---|---|---|
| | mg 100 g Fw$^{-1}$ | | | | (-) | |
| *Biochar (B)* | | | | | | |
| B− | 17.4 ± 0.9 | 72.3 ± 4.3 | 18.4 ± 1.1 | 90.8 ± 5.3 | 3.9 ± 0.0 b | 5.2 ± 0.1 |
| B+ | 16.3 ± 0.9 | 66.1 ± 4.4 | 16.2 ± 1.1 | 82.4 ± 5.5 | 4.1 ± 0.0 a | 5.0 ± 0.1 |
| *Compost (C)* | | | | | | |
| C− | 14.9 ± 0.5 b | 57.9 ± 1.5 b | 14.6 ± 0.2 b | 72.5 ± 1.6 b | 4.0 ± 0.1 | 4.9 ± 0.1 b |
| C$_{OP}$+ | 15.7 ± 0.5 ab | 65.6 ± 2.9 ab | 16.5 ± 0.6 ab | 82.1 ± 3.5 ab | 4.0 ± 0.0 | 5.2 ± 0.1 a |
| C$_W$+ | 17.7 ± 1.4 ab | 73.8 ± 6.2 ab | 18.6 ± 1.7 ab | 92.3 ± 7.9 ab | 4.0 ± 0.0 | 5.1 ± 0.1 ab |
| C$_D$+ | 19.1 ± 1.2 a | 79.5 ± 5.8 a | 19.7 ± 1.5 a | 99.3 ± 7.4 a | 4.0 ± 0.0 | 5.1 ± 0.1 ab |
| *Biochar x Compost* | | | | | | |
| B− C− | 14.1 ± 0.7 | 55.9 ± 2.5 | 14.5 ± 0.3 | 70.4 ± 2.7 | 3.8 ± 0.1 | 5.0 ± 0.1 |
| B+ C− | 15.7 ± 0.3 | 60.0 ± 1.0 | 14.6 ± 0.3 | 74.7 ± 1.1 | 4.1 ± 0.1 | 4.8 ± 0.0 |
| B− C$_{OP}$+ | 15.1 ± 0.4 | 63.8 ± 1.0 | 16.4 ± 0.2 | 80.2 ± 1.1 | 3.9 ± 0.1 | 5.3 ± 0.1 |
| B+ C$_{OP}$+ | 16.3 ± 0.4 | 67.5 ± 1.2 | 16.6 ± 0.3 | 84.0 ± 1.2 | 4.1 ± 0.1 | 5.1 ± 0.1 |
| B− C$_W$+ | 19.6 ± 0.4 | 81.0 ± 1.0 | 20.9 ± 0.3 | 102.0 ± 1.1 | 3.9 ± 0.1 | 5.2 ± 0.1 |
| B+ C$_W$+ | 15.9 ± 0.4 | 66.5 ± 2.1 | 16.2 ± 0.5 | 82.7 ± 2.5 | 4.1 ± 0.1 | 5.2 ± 0.2 |
| B− C$_D$+ | 21.0 ± 0.5 | 88.6 ± 1.4 | 21.9 ± 0.3 | 110.5 ± 1.5 | 4.1 ± 0.1 | 5.2 ± 0.1 |
| B+ C$_D$+ | 17.2 ± 0.3 | 70.5 ± 2.4 | 17.6 ± 0.6 | 88.0 ± 3.0 | 4.0 ± 0.1 | 5.1 ± 0.2 |
| *Significance* | | | | | | |
| B | ns | ns | ns | ns | ** | ns |
| C | ** | ** | * | ** | ns | * |
| B x C | ns | ns | ns | ns | ns | ns |

Values are means (*n* = 4) ± standard errors. In each column, means followed by the same letters are not significantly different (*p* ≤ 0.05; Tukey's test). *, F test significant at *p* ≤ 0.05; **, F test significant at *p* ≤ 0.01; ns, not significant. B, biochar; C, compost; C$_{OP}$, compost from olive pomace; C$_W$, vermicompost; C$_D$, compost from cattle anaerobic digestate; − and +, without and with, respectively.

The SPAD meter values detected at the first and the second leaf cut on the youngest fully expanded leaf (SPADleaf) and on the whole plant leaves (SPADplant) are reported in Table 5.

At the first leaf cut (Table 5), as already observed for pigment content, SPADleaf values resulted significantly influenced only by compost (*p* ≤ 0.05). In particular, C$_W$+ treated plants showed a higher SPADleaf value than C− (without compost), accounting for a 23% increase. At the same leaf cut, neither the main effect of the two factors, biochar and compost, nor the interactive effect of biochar x compost, resulted statistically significant on SPADplant values.

In addition, at the second leaf cut (Table 5), both SPADleaf and SPADplant resulted significantly influenced only by compost (*p* ≤ 0.05, *p* ≤ 0.01 for SPADleaf and SPADplant, respectively). Plants grown on C$_D$+ treated soil accounted for a 16% higher SPADleaf and 17% higher SPADplant value than C− (without compost).

### 3.3. Total Nitrogen and Nitrate Leaf Content

Total nitrogen (total N) content of leaves was significantly affected by biochar (*p* ≤ 0.001) and compost (*p* ≤ 0.001), as well as by their interaction (*p* ≤ 0.001), both at the first and the second leaf cut, as shown in Table 6.

**Table 5.** Effect of biochar, compost and biochar x compost interaction on SPAD meter values relatively to the youngest fully expanded leaf (SPADleaf) and the whole plant leaves (SPADplant), at the first and the second cut Swiss chard.

| Experimental Factor | First Cut | | Second Cut | |
|---|---|---|---|---|
| | **SPADleaf** | **SPADplant** | **SPADleaf** | **SPADplant** |
| | **(-)** | | **(-)** | |
| *Biochar (B)* | | | | |
| **B−** | $38.4 \pm 1.7$ | $29.1 \pm 1.0$ | $39.5 \pm 1.1$ | $36.0 \pm 0.9$ |
| **B+** | $36.4 \pm 1.5$ | $29.0 \pm 0.9$ | $37.7 \pm 0.9$ | $35.0 \pm 0.8$ |
| *Compost (C)* | | | | |
| **C−** | $32.8 \pm 1.7$ b | $26.2 \pm 1.4$ | $35.6 \pm 1.7$ b | $32.5 \pm 1.3$ b |
| **$C_{OP}$+** | $36.9 \pm 1.7$ ab | $29.2 \pm 1.0$ | $38.2 \pm 1.0$ ab | $35.2 \pm 0.6$ ab |
| **$C_W$+** | $40.4 \pm 1.6$ a | $30.7 \pm 0.8$ | $39.4 \pm 0.6$ ab | $35.9 \pm 0.7$ ab |
| **$C_D$+** | $39.4 \pm 2.2$ ab | $30.2 \pm 1.0$ | $41.3 \pm 1.2$ a | $38.1 \pm 1.1$ a |
| *Biochar x Compost* | | | | |
| **B− C−** | $32.8 \pm 2.9$ | $25.9 \pm 2.6$ | $36.0 \pm 3.5$ | $32.8 \pm 2.6$ |
| **B+ C−** | $32.9 \pm 2.4$ | $26.6 \pm 1.5$ | $35.3 \pm 1.1$ | $32.1 \pm 0.9$ |
| **B− $C_{OP}$+** | $40.0 \pm 2.2$ | $29.4 \pm 2.2$ | $39.3 \pm 2.5$ | $35.6 \pm 1.8$ |
| **B+ $C_{OP}$+** | $33.7 \pm 1.7$ | $28.9 \pm 1.3$ | $37.1 \pm 0.7$ | $34.8 \pm 0.9$ |
| **B− $C_W$+** | $39.4 \pm 2.3$ | $30.6 \pm 1.8$ | $39.8 \pm 2.5$ | $35.9 \pm 2.0$ |
| **B+ $C_W$+** | $41.4 \pm 0.9$ | $30.8 \pm 1.1$ | $39.1 \pm 1.7$ | $35.9 \pm 1.3$ |
| **B− $C_D$+** | $41.2 \pm 2.1$ | $30.6 \pm 1.5$ | $43.0 \pm 2.5$ | $39.5 \pm 2.0$ |
| **B+ $C_D$+** | $37.7 \pm 2.5$ | $29.8 \pm 2.0$ | $39.5 \pm 1.6$ | $36.7 \pm 1.4$ |
| *Significance* | | | | |
| **B** | ns | ns | ns | ns |
| **C** | * | ns | * | ** |
| **B x C** | ns | ns | ns | ns |

Values are means ($n = 4$) $\pm$ standard errors. In each column, means followed by the same letters are not significantly different ($p \leq 0.05$; Tukey's test). *, F test significant at $p \leq 0.05$; **, F test significant at $p \leq 0.01$; ns, not significant. B, biochar; C, compost; $C_{OP}$, compost from olive pomace; $C_W$, vermicompost; $C_D$, compost from cattle anaerobic digestate; − and +, without and with, respectively.

At the first cut (Table 6), total N leaf content was higher in plants grown on soil added with biochar (B+) than without biochar (B−), with an increase of 4%. A higher value of total N was also detected on $C_W$+ treated plants than $C_{OP}$+, $C_D$+, and C− (without compost). In particular, an increase of total N content by 56% was observed on $C_W$+ treated plants in comparison with C−. The highest total N leaf content was showed by plants grown on B− $C_W$+ treatment (without biochar and with vermicompost) and the lowest on the control B− C−, accounting for 157% increase.

Similar results were obtained at the second leaf cut (Table 6). When soil was added with biochar (B+), the total N content of leaves was higher than without biochar (B−), accounting for a 7% increase. Again, higher total N content was measured in $C_W$+ treated plants than $C_{OP}$+, $C_D$+, and C− (without compost). In particular, an increase of total N equal to 49% was found in $C_W$+ treated plants compared with C−. Total N content reached the highest value in plants grown on B+ $C_W$+ treatment, showing an increase of 128% in comparison to the control B− C−.

As already observed for total N, the nitrate ($NO_3^-$) leaf content was also influenced by biochar ($p \leq 0.001$) and compost ($p \leq 0.001$), as well as by their interaction ($p \leq 0.001$), both at the first and the second leaf cut (Table 6). At the first leaf cut, soil amendment with biochar (B+) resulted in a lower $NO_3^-$ leaf content than without biochar (B−), with a decrease of 20%. Moreover, plants grown on $C_{OP}$+ treated soil showed higher $NO_3^-$ leaf content than $C_W$+, C−and $C_D$+ that did not differ each other. In comparison with C−, the increase of $NO_3^-$ leaf content in $C_{OP}$+ treated plants was equal to 33%. The highest $NO_3^-$ leaf content was detected in plants grown on soil treated without biochar and with

compost from olive pomace (B–$C_{OP}$+), accounting for a 121% increase in comparison with the control (B–C–).

**Table 6.** Effect of biochar, compost and biochar x compost interaction on total nitrogen (total N) and nitrate ($NO_3^-$) leaf content, at the first and second cut of Swiss chard.

| Experimental Factor | First Cut | | Second Cut | |
|---|---|---|---|---|
| | Total N % of Dw | $NO_3^-$ mg kg $^{-1}$ Fw | Total N % of Dw | $NO_3^-$ mg kg $^{-1}$ Fw |
| *Biochar (B)* | | | | |
| B– | 1.9 ± 0.2 b | 249.5 ± 18.8 a | 1.6 ± 0.1 b | 261.0 ± 17.3 a |
| B+ | 2.0 ± 0.1 a | 199.0 ± 17.8 b | 1.7 ± 0.1 a | 225.1 ± 16.1 b |
| *Compost (C)* | | | | |
| C– | 1.6 ± 0.2 d | 206.2 ± 16.9 b | 1.3 ± 0.2 d | 218.1 ± 21.1 c |
| $C_{OP}$+ | 2.0 ± 0.1 b | 273.7 ± 33.3 a | 1.7 ± 0.0 b | 274.4 ± 27.9 a |
| $C_W$+ | 2.5 ± 0.2 a | 215.2 ± 9.6 b | 1.9 ± 0.0 a | 242.2 ± 7.3 b |
| $C_D$+ | 1.9 ± 0.0 c | 201.9 ± 13.4 b | 1.6 ± 0.1 c | 237.4 ± 15.1 b |
| *Biochar x Compost* | | | | |
| B– C– | 1.1 ± 0.0 f | 163.1 ± 8.5 c | 0.9 ± 0.0 f | 163.8 ± 9.8 e |
| B+ C– | 2.1 ± 0.0 c | 249.2 ± 4.6 b | 1.7 ± 0.0 d | 272.5 ± 2.2 b |
| B– $C_{OP}$+ | 1.8 ± 0.2 e | 360.9 ± 19.3 a | 1.7 ± 0.2 d | 347.6 ± 28.8 a |
| B+ $C_{OP}$+ | 2.2 ± 0.1 b | 186.6 ± 24.1 c | 1.8 ± 0.0 c | 201.2 ± 15.6 d |
| B– $C_W$+ | 2.9 ± 0.3 a | 238.6 ± 25.7 b | 1.9 ± 0.2 b | 257.3 ± 33.9 b |
| B+ $C_W$+ | 2.1 ± 0.1 c | 191.9 ± 32.6 c | 2.0 ± 0.0 a | 227.2 ± 19.1 cd |
| B– $C_D$+ | 2.0 ± 0.2 c | 235.3 ± 23.6 b | 1.9 ± 0.2 b | 275.2 ± 32.2 b |
| B+ $C_D$+ | 1.9 ± 0.1 d | 168.5 ± 25.3 c | 1.3 ± 0.0 e | 199.5 ± 17.1 d |
| *Significance* | | | | |
| B | *** | *** | *** | *** |
| C | *** | *** | *** | *** |
| B x C | *** | *** | *** | *** |

Values are means ($n$ = 4) ± standard errors. In each column, means followed by the same letters are not significantly different ($p \leq 0.05$; Tukey's test). ***, F test significant at $p \leq 0.001$; ns, not significant. B, biochar; C, compost; $C_{OP}$, compost from olive pomace; $C_W$, vermicompost; $C_D$, compost from cattle anaerobic digestate; – and +, without and with, respectively.

At the second leaf cut (Table 6), plants grown on soil amended with biochar (B+) again resulted in a lower $NO_3^-$ leaf content than without biochar (B–), accounting for a 14% decrease. The $NO_3^-$ leaf content was higher in plants grown on $C_{OP}$+ treatment than $C_W$+ and $C_D$+ that did not differ each other, and C– (without compost). The highest $NO_3^-$ leaf content was again detected in B–$C_{OP}$ treatment and the lowest in B–C–, with a 112% increase.

## 4. Discussion

### 4.1. Plant Growth

The findings of our study clearly showed a negative effect of biochar soil addition on plant height, leaf area, and leaf fresh weight of Swiss chard. On the contrary, soil amendment with composts, particularly those from animal wastes, such as vermicompost from cattle manure and compost from cattle anaerobic digestate, effectively enhanced plant growth. These results are consistent with our previous study [23] confirming, on the one hand, the non-beneficial effect of the tested biochar on Swiss chard growth performance and, on the other hand, the fertilizing value of the considered composts. This finding allows to speculate that Swiss chard could be usefully oriented toward the cultivation with organic amendments to replace chemical fertilizers, while preserving the environmental sustainability of crop production.

With regard to the positive effect of vermicompost, our results are also in agreement with other studies. Indeed, this organic amendment has been reported to enhance the

growth of a wide range of plant species including cereals, legumes, horticultural and fruit crops, aromatic and medicinal species, ornamentals, and forestry plants [36,37]. Increasing the Swiss chard growth parameters observed in our experiment could be attributed to the higher amounts of available mineral N and P in the soil amended with vermicompost. According to Lazcano and Dominguez [36], vermicompost constitutes a source of macro ($NO_3$, $PO_4$, Ca, K, Mg, S) and micronutrients in forms readily available to plants or gradually released through organic matter mineralization, thus exhibiting similar effects as inorganic fertilizers. In addition, the authors described that vermicompost is a finely-divided material able to improve soil porosity and water holding capacity thus promoting plant rooting, and represents a microbiologically active organic manure containing growth regulating hormones. These two physical and biological mechanisms directly influencing plant growth could also explain why, in our case, soil treatments with vermicompost determined higher plant height, leaf area and leaf fresh weight. Swiss chard responded positively also to compost from cattle anaerobic digestate and this result was likely due to a greater release of mineral elements for plant nutrition, mostly N, in the treated soil. In addition, Suvendu et al. [38] observed that soil amendment with compost from cattle manure increased biomass and grain yield of a paddy rice crop, due to the improvement of plant nutrients (particularly C, N, and P) availability in the soil. Soil treatment with compost from olive pomace resulted in a lower Swiss chard growth than the other two composts, probably due to the higher C/N ratio (see Table 2) that determined a higher N immobilization in soil, a higher N competition in plants and soil microorganisms and a lower N availability for plant growth. According to this hypothesis are the results of Morra et al. [39], who found a higher quickness of buffalo manure amendment than biowaste and olive pomace compost in supplying mineral nitrogen to a rocket crop. Our hypothesis is also supported by previous findings of Garcia-Ruiz et al. [40]. The authors reported an N immobilization, with reduced nitrate losses from the soil, in the short-term (3–12 months) olive pomace compost decomposition and a total N amount and N easily mineralized pool increase, in the long term (15 years).

Other results of the present study concern the interactive effect of biochar and composts on Swiss chard plant. In particular, our findings did not support the hypothesis of an enhanced growth response following soil addition with biochar-compost mixtures. Contrary to the supposed results, mixing the tested biochar with the three considered compost types did not significantly affect any of the measured quantitative parameters. These results are in agreement with other authors. Indeed, Agegnehu et al. [41] observed that biochar, compost, and particularly their combination did not result in significant differences of peanut seed yield and total plant biomass. Moreover, Trupiano et al. [42] observed that compost and biochar positively affected lettuce growth and yield, but their combination did not exert any synergic influence. In our study, the observed growth response of Swiss chard to soil addition with biochar-compost mixtures leads to hypothesize that biochar particles, due to sorption ability, have retained some mineral elements and particularly mineralized N and available P. Consequently, the nutrients availability to plants in soil treated with biochar-compost mixtures was likely decreased, leading to a lower growth response, as also reported by Seehausen et al. [43]. In addition, the usually high biochar C/N ratio could lead to an immobilization of N [44] and particularly of $NO_3^-$-N [45], in the biochar amended soils, thus reducing N supply and limiting plant growth and productivity. This likely occurred in our experiment, due to the high C/N ratio of the tested biochar (see Table 2).

*4.2. Pigment Leaf Content*

Carotenoid and chlorophyll contents of Swiss chard leaves were increased by soil treatment with compost from cattle anaerobic digestate (79%), crop residues (11%), and wheat straw (10%). This is an interesting finding since it allows to speculate that the higher leaf pigment content could greatly increase dietary nutritional contributions

to the human diet when consuming Swiss chard grown on soil added with this type of organic amendment.

Carotenoid content was higher than that found on Swiss chard by Ivanović et al. [46], who reported values in the range from 4.4 to 13.4 mg 100 g $Fw^{-1}$, and by Reif et al. [47], who found a total content of carotenoids (as a sum of lutein and β-carotene) from 3.7 to 9.6 mg 100 g $Fw^{-1}$. The leaf chlorophyll content was lower than that reported by Žnidarčič et al. [48] for some other leafy vegetables similar to Swiss chard, such as garden rocket (359.6 mg 100 g $Fw^{-1}$), wild rocket (303.2 mg 100 g $Fw^{-1}$), and dandelion (248.2 mg 100 g $Fw^{-1}$). The differences in pigment concentrations among these studies may be due to different physiological, genetic, and biochemical characteristics of plant species/cultivars used, environmental factors and cultivation conditions (light, temperature, and fertilization), as well as to differences in analytical methodologies [48]. As already discussed in our previous paper [23], the significant effect of compost from cattle anaerobic digestate on leaf pigment content (carotenoids and chlorophyll) was likely due to an improved plant nutritional status and particularly to the higher N availability in soil treated with this type of compost. In this regard, Miceli and Miceli [19] and Ivanovic et al. [46] found that higher N fertilization increases chlorophyll a and chlorophyll b synthesis in Swiss chard.

In our study, chlorophyll a to chlorophyll b ratio showed similar values to those reported in literature for other green leafy tissues [49]. Moreover, the chlorophyll a to chlorophyll b ratio significantly increased with biochar soil application likely due to a limited N availability for plant nutrition, as already discussed in the previous section, in the treated soil. In addition, Kitajima and Hogan [49] observed increases of chlorophyll a to chlorophyll b ratio in four tropical plant leaves in response to N limitation. The total chlorophyll to carotenoids ratio increased when soil was amended with compost from olive pomace, likely due to a higher $NO_3^-$-N than NH4+-N availability for plant nutrition. In this regard, Barickman and Kopsell [50] reported that N form and ratio can influence leaf pigment content in Swiss chard and found significant increases in carotenoids and chlorophyll content in response to decreasing ratios of NH4+-N to $NO_3^-$-N in the nutrient solution.

### 4.3. Total Nitrogen and Nitrate Leaf Content

Still considering our results, soil addition with biochar induced a higher total N leaf content. This seems a contradictory result if considering the lower height, leaf area, and leaf fresh weight measured on plants treated with biochar (see Table 3). We hypothesized that the higher total N leaf content was due to a higher $NH_4^+$ availability in the soil that was used as a higher rate of N source than $NO_3^-$. Indeed, at higher NO3- concentration than $NH_4^+$, plants growth is enhanced; on the contrary, when $NH_4^+$ is mainly used as N source, plants exhibit a stunted growth with decreased leaf area [50]. Our hypothesis seems to be consistent also with the lower $NO_3^-$ leaf content of Swiss chard plant grown on soil added with biochar in comparison with treatments not including biochar (see Table 6). In addition, soil addition with compost increased the total N content of Swiss chard leaves. Particularly, the total N leaf content was higher in soil treated with vermicompost from cattle manure than the two composts from olive pomace and cattle anaerobic digestate, respectively. This was likely due to a higher N availability for plant uptake, as also showed by the higher plant growth. In our study, N leaf content observed on these plants was higher than N leaf content found by Hernández et al. [51] in lettuce plants also treated with a vermicompost from cattle manure, having a similar N content (1.6%) and applied to the soil in order to provide a similar N amount (278 kg N $ha^{-1}$) to vermicompost used in our experiment. Such differences could be likely related to the dynamics of nutrient release in the soil after incorporating organic fertilizers, which depend not only on the total mineral element amount, but largely on the amendment characteristics [52]. Among the latter, the stability and maturity of organic matter, as well as the compost physico-chemical properties, highly influence the N availability. Moreover, the environmental conditions, such as soil type and climate, can affect the N dynamic after compost soil addition, making difficult the comparison of different experiments [53,54].

Considering the results of our study about $NO_3^-$ content of Swiss chard leaves, these were in the range reported by EFSA (European Food Safety Authority) [55] and by Colla et al. [17]. $NO_3^-$ leaf content always resulted well below the maximum levels set by the European Commission (Regulations No. 1881/2006 and 1258/2011) for the following leafy vegetables: fresh spinach (3500 mg kg$^{-1}$ Fw); preserved, deep-frozen or frozen spinach (2000 mg kg$^{-1}$ Fw); fresh lettuce (3000–5000 mg kg$^{-1}$ Fw); Iceberg type lettuce (2000–2500 mg kg$^{-1}$ Fw); salad and wild rocket. (6000–7000 mg kg$^{-1}$ Fw). Among the several factors influencing the accumulation of nitrate in leafy, crop cultivation system plays an important role, and, in this regard, there are consistent findings that show lower nitrate concentration in organic-amended vegetable crops [56]. The results of our study are in agreement with the findings of Raigon et al. [57] who observed a nitrate content below 400 mg kg$^{-1}$ Fw for Swiss chard plants cultivated under organic system farming. In our study, $NO_3^-$ content was higher in Swiss chard plants grown on soil treated with compost from olive pomace likely due to the higher $NO_3^-$-N availability than $NH_4^+$-N. In addition, Barcelos et al. [58] and Conesa et al. [59] reported that spinach accumulated more nitrate when grown with a nutrient solution characterized by a high $NO_3^-$-N to $NH_4^+$-N ratio. Moreover, a nitrate uptake exceeding the assimilation capacity (nitrate reduction in ammonium, which is fixed into the amino acids glutamine and glutamate) likely occurred in the course of the Swiss chard growth cycles, leading to nitrate leaf accumulation [17]. The lower height, leaf area, and leaf fresh weight of Swiss chard plants grown on soil treated with compost from olive pomace could support our hypothesis of nitrate plant uptake exceeding nitrate reduction. On the contrary, biochar addition to the soil reduced $NO_3^-$ content in Swiss chard leaves, likely due to the occurrence of a nitrate sorption on biochar surface. Several authors showed that biochar is effective in retaining nitrates in the soil [60–65] and attributed this ability to the high pyrolysis temperature (>600 °C) at which the biochar is obtained [66]. According to the literature, the biochar nitrate retention we hypothesized in our experiment could be explained considering the high temperature (650 °C) applied to vine pruning residues for biochar production. The biochar-nitrate retention has been reported as the fundamental process by which biochar promotes plant growth, as nitrate release from biochar amended soil is more slowly [63] than non-biochar-amended soils, thus preventing nitrate leaching and providing nitrate to plants over a longer period of time. Nevertheless, in our experimental conditions, the hypothesized nitrate retention by biochar resulted in a decrease of nitrate availability for plant uptake, due to the short growth cycle of Swiss chard plants and the biochar benefits that could be achieved over time.

## 5. Conclusions

The present study investigated the effect of soil addition with biochar from vine pruning residues, respectively, mixed with a compost from olive pomace, a vermicompost from cattle manure, and a compost from cattle anaerobic digestate (79%), crop residues (11%), and wheat straw (10%) on the quantitative and qualitative response of Swiss chard.

In a factorial experiment, the main effect of biochar resulted in a lower plant growth. Moreover, biochar soil application did not influence the qualitative parameters, except for total N leaf content that was enhanced and $NO_3^-$ content that, on the contrary, was reduced. The main effect of composts, particularly vermicompost from cattle manure and compost from cattle anaerobic digestate, was significant on plant height, leaf area, leaf fresh weight, carotenoid, and chlorophyll leaf contents. The positive effect of vermicompost also reflected in the higher leaf total N content, thus suggesting the adequate fertilizing value of the tested product. Leaf $NO_3^-$ content, although increased by compost from olive pomace, was always well below the $NO_3^-$ maximum levels in leafy vegetables, set by European Commission to avoid adverse human health effects after raw vegetable consumption. The interactive effect of biochar and compost did not produce the expected results. Indeed, biochar-compost mixtures did not affect quantitative nor qualitative plant response. A low

nutrient status, likely due to a blocking mineral elements in the soil by biochar, could have caused such effects.

In summary, the results of our experiment suggest that soil addition with compost represent the best option to increase plant biomass and leaf concentration of health-promoting compounds in short-term Swiss chard crop. Considering the biochar long-term stability in the soil and the biochar benefits that may be achieved over time, more long-term studies are required to better understand the interactive effect of biochar and compost within a sustainable production system aimed to increase yield and quality of horticultural crops.

**Author Contributions:** Conceptualization, A.L. and A.R.R.; methodology, A.L. and A.R.R.; formal analysis, A.L.; investigation, A.L. and A.R.R.; data curation, A.L. and A.R.R.; writing—original draft preparation, A.L.; writing—review and editing, A.L.; visualization, A.L. and A.R.R.; supervision, A.L. and A.R.R.; funding acquisition, A.R.R. All authors have read and agreed to the published version of the manuscript.

**Funding:** This research was carried out in the framework of the project "Smart Basilicata", which was approved by the Italian Ministry of Education, University and Research (Notice MIUR n.84/2012, PON 2007–2013 of 2 March 2012) and was funded by the Cohesion Fund 2007–2013 of the Basilicata Regional Authority.

**Institutional Review Board Statement:** Not applicable.

**Informed Consent Statement:** Not applicable.

**Data Availability Statement:** The data presented in this study are available on request from the corresponding author.

**Acknowledgments:** The authors are grateful to Susanna De Maria, Giuseppe Mercurio, and Antonio Pisani for the technical assistance in the greenhouse experiment and the laboratory analyses.

**Conflicts of Interest:** The authors declare no conflict of interest. The funders had no role in the design of the study; in the collection, analyses, or interpretation of data; in the writing of the manuscript, or in the decision to publish the results.

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
