# Peer review of "Quanti-Qualitative Response of Swiss Chard (Beta vulgaris L. var. cycla) to Soil Amendment with Biochar-Compost Mixtures"

_agronomy, doi:10.3390/agronomy11020307_

Round 1

Reviewer 1 Report

The aim of the study in „Quanti-qualitative Response of Swiss chard (Beta vulgaris L. var. cycla) to Soil Amendment with Biochar-Compost Mixtures” was to test if mixing biochar with composts could be a more powerful strategy for enhancing biochar effect on plant growth and qualitative characteristics of product, in a perspective of sustainable agriculture.

The article is very interesting. Congratulations – statistical research is very important in experiments, however, it should have some minor improvements.

To improve aesthetics, create a "units" column in tables.

I believe that the article should be revised and considered for adoption as amended.

The manuscript should be supplemented with statistical analyzes, which is very important in scientific research.

Standard deviations should be added to the results (I assume that the researchers know that the correct performance of the tests requires at least three repetitions).

In my opinion, authors should add the standard deviations in the results listed in the text (e.g. ln 305, 320, 334, 340...);

Ln 233 Statistical analysis is very important in the experiments and congratulations they are included. However, it could be extended e. g. the hypothesis on distribution of each analyzed variable can be verified with a Shapiro-Wilk test, Variance homogeneity in groups can be checked with Levene's test.

If they have been tested, complete the methodology.

Besides, I think it will interest readers.

Reviewer 2 Report

Dear editor,

The authors of the manuscript entitled “Quanti-qualitative Response of Swiss chard (Beta vulgaris L. var. cycla) to Soil Amendment with Biochar-Compost Mixtures” have investigated the effect of a combined application of biochar and compost on Swiss chard. This research presents interesting results, however, there are some points of concerns:

  • In recent years, there have been many papers on the effect of biochar alone or in combination with compost on the growth of different plants. Unfortunately, there is no novelty in this research. Moreover, the research was limited to a pot experiment. It would have been better to combine it with a field trial or a more large scale test.
  • This experiment has been performed only once. Conclusions are not supported by results from repeated experiments and therefore reproducibility of results is not illustrated. Drawing conclusions based upon only one short-term test is not very logical.
  • Authors have determined soil and amendments’ characteristics only at the start of the experiment; however, soil chemical characteristics after the addition of amendments (e.g., at the end of the experiment) should have been determined.
  • In the section of ‘Discussion’, authors could not discuss their obtained results based on their own data. Indeed, they repeatedly referred to the results of other authors. In my opinion, this is mostly due to a lack of data (e.g., data on soil chemical and/or biological analysis) which could have helped them explain their findings.
  • In ‘Discussion’, the authors did not mention anything about the possible effect of the pyrolysis process or the biochar food stock on the performance of biochar.
  • The paper needs to be revised to improve some English grammatical mistakes.

Round 2

Reviewer 2 Report

Dear editor,

The authors have improved the English language and typos in the text. However, there are still some grammatical mistakes in the parts that have been added in the revised manuscript, e.g., line 400, "These findings" instead of "This finding". 

Author Response

Dear Reviewer,

We apologize for the mistake. According to your kind suggestion, we changed “This findings” with “This finding” (Line 445 of the revised manuscript version).

We further checked the parts added in the manuscript for any other grammatical mistakes and, if necessary, corrected them.

Thank you for your further contribution in improving the quality of the manuscript.

Best regards,

The corresponding author

Angela Libutti